# Concordance among patients and physicians about their ideal of autonomy impacts the patient-doctor relationship: A cross-sectional study of Mexican patients with rheumatic diseases

**Virginia Pascual-Ramos**[1☯]*, **Irazú Contreras-Yáñez**[1☯], **Ana Belén Ortiz-Haro**[1☯],
**Christiaan Molewijk Albert**[2☯], **Gregorio Tomás Obrador**[3☯], **Evandro Agazzi**[4☯]

1 Department of Immunology and Rheumatology, Instituto Nacional de Ciencias Médicas y Nutrición Salvador Zubirán, Mexico, Mexico, 2 Department Medical Humanities, Amsterdam, UMC, VUmc, Netherlands, 3 Universidad Panamericana, Interdisciplinary Center of Bioethics and School of Medicine, Mexico, Mexico, 4 Universidad Panamericana Interdisciplinary Center of Bioethics, Mexico, Mexico

☯ These authors contributed equally to this work.
* virtichu@gmail.com

## Abstract

### Introduction

In patient-doctor interaction both parties play a role. Primary objective was to determine if the concordance among rheumatologists and their patients of their ideal of autonomy was associated with a better patient-doctor relationship. Secondary objective was to describe factors associated to a patient paternalistic ideal of autonomy (PPIA).

### Materials and methods

This cross-sectional study had 3 steps. Step-1 consisted in translation/cultural local adaption of Ideal Patient Autonomy Scale (IPAS), a 14-items Dutch questionnaire. Step-2 consisted of IPAS validity and reliability in 201 outpatients. Step-3 consisted of the application of IPAS and the patient-doctor relationship questionnaire (PDRQ) to 601 outpatients with a medical encounter, and of IPAS to the 21 attending rheumatologists. Each patient-physician encounter was classified into with/without concordance in the ideal of autonomy and PRDQ scores were compared (Man Whitney U test). Regression analysis was used for associations.

### Results

Step-1 followed ISPOR task force recommendations. Patients from Step-2 and Step-3 were representative outpatients with rheumatic diseases. IPAS structure underwent a modification; the 14 items were redistributed into four subscales, further combined into PPIA vs. patient-centered autonomy ideal. IPAS was valid and reliable. There were 497 patients with a preferred ideal of autonomy, primarily (84.9%) PPIA. There were 363 patient-doctor

**Data Availability Statement:** All relevant data are within the manuscript and its Supporting Information files.

**Funding:** The authors received no specific funding for this work.

**Competing interests:** The authors have declared that no competing interests exist.

encounters with concordance in the autonomy ideal and their PDRQ-9 scores were higher. Religious beliefs and higher PDRQ-9 item 8 score ("I feel pleased with my doctor´s treatment") were associated to a PPIA.

## Conclusions

Concordance of autonomy ideal among patients and their rheumatologists positively impacts on the patient-doctor relationship.

## Introduction

Rheumatologists are considered the primary care physicians of patients with rheumatic diseases, which are integrated by a complex group of chronic musculoskeletal conditions, primarily characterized by musculoskeletal pain that substantially contributes to a deteriorated patient´s quality of life [1]. In recent years, patient-centered care has been recognized as necessary to complement the evidence-based health care imposed in the last decades, and has been proposed as the optimal conceptual model of care for patients affected by rheumatic diseases, where patients are required to actively implement and maintain changes in their health-related behavior, in addition to strongly adhering to treatment strategies [2]. Breen et al. [3] defined patient-centric care as the clinical treatment provided by medical professionals that focuses on respecting and attending to patient preferences, desires, and values; respect to (a particular concept of) the patient´s autonomy is implicit in such a definition.

In terms of health, patient autonomy means the freedom to make responsible decisions that affect their own health. This concept of autonomy corresponds to the interpretation of liberal and individualist autonomy defined by Beauchamp and Childress [4], which has been adopted by the current ethical discourse of the medical literature. However, the ethics literature highlights other normative connotations of patient autonomy, and some are potentially of great use to describe and prescribe current medical practices according to the patient-centric care model [5]. Examples are autonomy as a critical reflection, as proposed by Dworkin [6], according to which a patient may decide to leave all decisions to the physician; this approach offers room for conscious submission to an external authority and is defined as paternalism with permission. Also, a different conceptualization considers autonomy as identification, which priorities the process of patient identification with the action, and not so much to make the decision him or herself [7].

In fact, despite the individual and liberal autonomy rhetoric among some bioethicists and members of the medical community, there is published evidence in non-rheumatologic [8–10] and rheumatologic fields [10–12] that patients want to be informed while they may not necessarily want to make decisions or challenge the physician´s authority; even more, in patients with rheumatic diseases, better information sharing improves patient satisfaction, compliance with treatment strategies, and health outcomes [13]. The discrepancies between patients´ desire for information and their desire to make decisions challenge the liberal and individualist concept of autonomy and are characteristic of patients with chronic diseases, and have been related to the "sick role" adopted by many patients [14]. In addition, power distance, which is defined as the degree to which people accept unequal power statuses, influences patient-doctor communication and the share-decision process, and may impact on outcomes attributed to that communication [15]. Accordingly, patients from a high-power-distance culture, which is the predominant culture in the Latin-American region, may expect the physician to take a

more authoritative approach during the medical encounter. Meanwhile, two studies performed in samples of ethnically diverse adult systemic lupus erythematosus (SLE) patients in the United States, have shown that more active patient-physician interactions have been associated with better outcomes [16, 17].

Finally, patient-doctor interaction, like other communicative encounters, is a process of mutual influence in which each participant´s behavior may constrain or facilitate the other´s response [18]. This mutual influence has been evident in patients with different chronic conditions [19], including patients with SLE [18] and rheumatoid arthritis (RA) [20]. Interestingly, in a study performed on 115 Japanese RA patients who were under the continuous care of eight rheumatologists, patient preferences for decision-making affected the association between reported participation in visit communication and the feeling of being understood by the attending rheumatologist [20]. Among patients who preferred autonomous decision-making, the likelihood of being understood was positively associated with the extent of reported participation in visit communication, whereas such a relationship was less evident among those with a lower preference for decision-making. Ultimately, in any patient-doctor relationship, whether patient centered or physician centered, both parties play a role and it is in this interaction that decisions are made, and the views of both are therefore relevant. It seems reasonable to assume a positive impact on the patient-doctor relationship when similar ideals for the share-decision process are shared by both parts.

With the above considerations in mind, the primary objective of the study was to determine if the concordance among rheumatologists and their patients of their ideal of autonomy was associated with a better (patient´s) perceived patient-doctor relationship in Mexican patients with rheumatic diseases. Secondary objectives were to describe the distribution of patients and physicians ideals of autonomy and the factors associated with the preference for a paternalistic ideal of autonomy.

## Patients and methods

### Setting and study population

This cross-sectional study was performed between July 2019 and February 2020 at the outpatient clinic of the Department of Immunology and Rheumatology, at a tertiary-care level, academic center for rheumatic diseases, located in Mexico City. The outpatient clinic is served by 11 rheumatologists and 10 trainees in rheumatology (assigned to the 2018–2020 training program). Around 5,000 patients currently attend the clinic and suffer from a variety of rheumatic diseases—the ten most frequently diagnosed are depicted in Supplementary Table 1 in S1 Appendix, (Please refer to the SI "Supplementary Table 1 in S1 Appendix. N˚ (%) of patients with at least one visit to the outpatient clinic with the 10 most frequent diagnosis specified", S1 Appendix), namely, SLE, RA, Sclerodermia, Systemic Vasculitis (SV), Primary Sjögren Syndrome (PSS), Spondyloarthropaties (SA), Inflammatory Miopathies (IM), Primary Anti-Phospholipid Syndrome (PAPS), Mixed Connective Tissue Disease (MCTD), and Adult Still disease.

During the study period, consecutive outpatients with a defined rheumatologic diagnosis (among those included in Supplementary Table 1 in S1 Appendix, but "other diagnosis") were invited to participate. Exclusion criteria included patients on palliative care, patients with Overlap Syndrome (but secondary Sjögren Syndrome) and patients with uncontrolled comorbid conditions.

In addition, all attending physicians, including trainees, were invited to participate.

## Study design

This cross-sectional study was performed in three phases.

*Phase 1* consisted of the translation and cultural adaptation to a Mexico version of the Ideal Patient Autonomy Scale (IPAS), a normative Dutch instrument that was developed to assess ideals of patient autonomy, from a broader perspective than that of liberal individualism [5]. The original instrument consisted of 14 items distributed into four sub-scales: "Doctor knows best," "Patient should decide," "Right to non-participation," and "Obligatory risk information" (Fig 1). Principles of good practice for the translation and cultural adaptation process for patient-reported-outcomes measures (PROMs) from the ISPOR task force were followed [21].

*Phase 2* consisted of the IPAS psychometric validation, which was performed after pilot testing to examine IPAS feasibility. Judgment experts determined IPAS's content validity. The validation group was integrated by seven rheumatologists and six professionals with master's degree in bioethics. Each expert independently rated each of the 14 items (from the Spanish version) according to the presence or absence of relevance, adequate wording, appropriate language, and meaning. Construct validity was evaluated using a factorial analysis. Criterion validity was examined according to the patient´s preferred decision-making role, to a question that evaluated patients' preferences for participation in treatment decision-making [9]. IPAS reliability was assessed with internal consistency and temporal stability, which was tested after the IPAS was applied to 50 patients, twice, within a 2±1 week interval.

*Phase 3* was designed to accomplish primary and secondary objectives and consisted of the application of the IPAS and the Spanish version of the Patient-Doctor Relationship Questionnaire 9 items (PDRQ-9) [22] to consecutive outpatients with rheumatic diseases. The

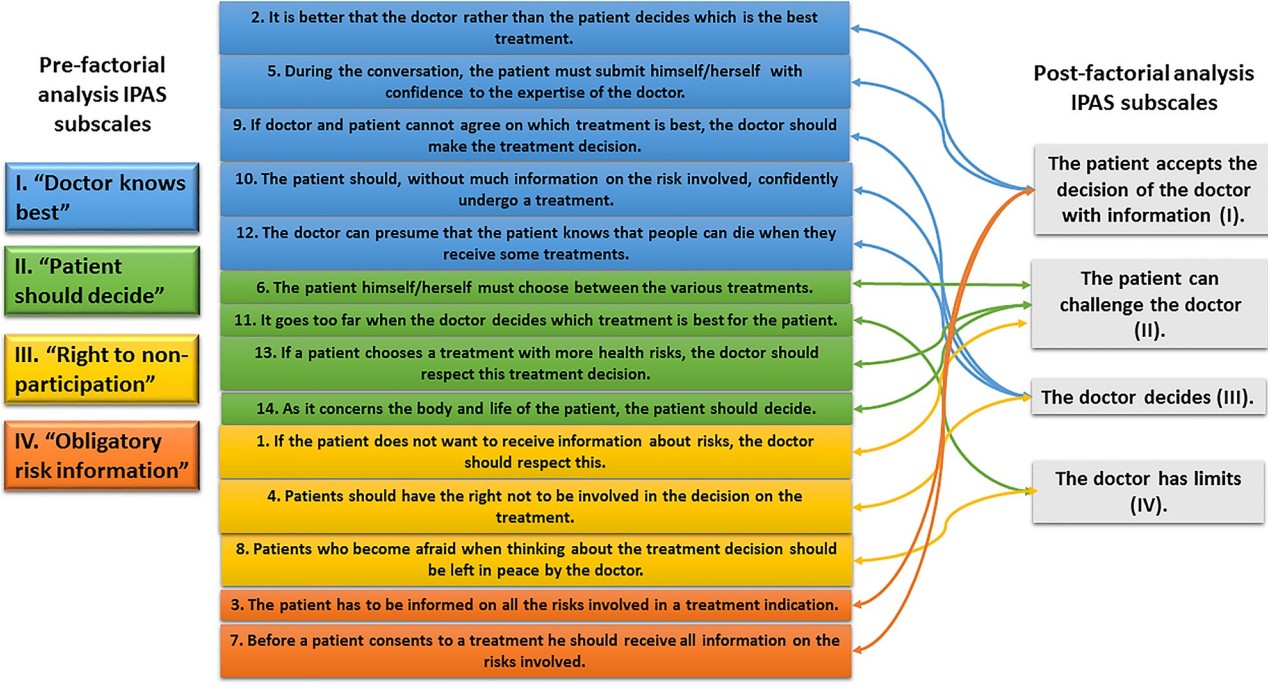

**Fig 1. IPAS structure pre and post factorial analysis.**

PDRQ-9 assesses the quality of the patient-doctor relationship experienced by the patient, through the quantification of the patient´s opinion regarding communication, satisfaction, and accessibility in dealing with the doctor and the treatment that followed. The questionnaire is based on a 5-point Likert scale from 1 (not at all appropriate) to 5 (totally appropriate); higher scores correlate with a better patient-doctor relationship. The IPAS was also applied to all attending physicians assigned to the patient´s care during the study period. For each patient-physician encounter, the patient's and the physician's preferred autonomy ideal was identified based on the IPAS sub-scale with the highest score [5], and accordingly, the patient-physician encounter was classified as with or without concordance in the ideal of autonomy; in addition, a PDRQ-9 score was assigned to each encounter. All patients additionally completed the Spanish version of the Health Assessment Questionnaire-Disability Index (HAQ-DI) to assess disability [23], of the Short Form 36-items (SF-36) to assess quality of life [24] and a visual analog scale (VAS) to assess pain. Finally, relevant sociodemographic variables (sex, years of age and of formal education, socioeconomic-level, religious beliefs, economic dependency, living with a partner, and access to the social security system), disease-related variables (disease duration, years of follow-up at the outpatient clinic, comorbid conditions and Charlson comorbidity score [25], participation in clinical trials, previous hospitalizations and number), and treatment-related variables (immunosuppressive treatment and number of immunosuppressive drugs/patient, and corticosteroid use) were obtained in standardized formats after a careful chart review and patient interview.

In all cases, questionnaires and scales were applied in a designated area for research purposes by personnel not involved in patient care.

## Description of sample and sample size calculation

Three different convenience samples of consecutive outpatients with a definite rheumatic disease, which was performed according to their attending rheumatologist criteria, were included. In the three samples, quotes were considered to represent the ten most frequent rheumatologic diagnoses (excluding the pilot testing sample). All patients were recruited from the outpatient clinic, where the patients' diagnoses have been coded and updated on a regular basis since 2003. In addition, 13 outpatients participated in phase 1.

The first sample (S-1) included 40 patients who were diagnosed with six of the ten most frequent rheumatologic diagnoses; the number of patients included followed the recommendations for pilot testing [26].

The second sample (S-2) included 201 patients, and was used for the IPAS validation process. The sample size was based on the methodological recommendations, which suggested a minimum of 50 patients for assessing construct validity, a minimum of 100 patients for assessing internal consistency, and five to ten patients for each item of the instrument [27]. After considering the additional published recommendations of at least 150 to 200 patients for factorial analysis [28], we decided to include at least 200 consecutive RA patients (fair sample size) in the final sample.

The third sample (S-3) included 601 patients. To achieve the primary objective, we estimated a sample size of 371 patients, and took into consideration that 80% of the patient-physician encounters with concordance in the ideal of autonomy would achieve the highest PDRQ-9 score vs. 65% of the encounters without concordance. We added 20%, due to the finding that only 82% of the patients from the phase 2 study had a predominant ideal of autonomy (according to the highest sub-scale score), and as a result 446 patients were included. We further explored the number of patients necessary to achieve the secondary objective. As 7,627 patients

were at the time already coded with a definite rheumatic diagnosis within the Department of Immunology and Rheumatology's database, we obtained a final sample size of (at least) 423 patients (after adding 20% for losses), with a 95% confidence level and 4% precision.

## Statistical analysis

Descriptive statistics were performed to estimate the frequencies and percentages for categorical variables or the mean and standard deviation (SD) for continuous variables, of the sociodemographic variables, the disease-related and treatment-related variables, and the patient-doctor relationship of the main sample.

Face and content validity by experts was examined with agreement percentages. Cronbach's α was used to assess the internal consistency of the questionnaire. For test-retest, intra-class correlation coefficients (ICC) and their 95% confidence intervals (CI) were calculated based on a single measurement, absolute-agreement, 2-way mixed-effects model. Cronbach's α, ICC, and 95% CI interpretations followed published recommendations [29, 30]. Construct validity was evaluated using exploratory factorial analysis (principal components) with Varimax rotation. Sampling adequacy was confirmed using the Kaiser-Mayer-Olkin (KMO) (appropriate value ≥0.5) measure, and the use of factorial analysis was supported by Bartlett's test of sphericity (significant value p<0.05). The number of factors was determined as the number of eigenvalues >1. For criterion validity, the distribution of patients' preferences for participation in treatment decision-making among groups of patients was defined based on their ideal of autonomy compared with the $X^2$ test.

Global PDRQ-9 scores and individual item scores, and the proportion of patients with the highest PDRQ-9 scores were compared between the patient-doctor encounters with concordance in the ideal of autonomy and their counterparts, using the Mann-Whitney U and $X^2$ tests, respectively. A simple linear regression analysis was used to investigate the relationship between patient-doctor concordance in their ideal of autonomy and global and individual items PDQR-9 scores.

Characteristics of the patients with an autonomy ideal paternalistic (corresponding to sub-scales I and III) were compared with those of patients with an autonomy ideal patient-centered (corresponding to sub-scales II and IV) using appropriate tests. Logistic multiple regression analysis were used to establish factors associated with a patient´s paternalistic ideal of autonomy, which was considered the dependent variable. The selection of the variables to be included was based on their statistical significance in the bivariate analysis (p≤0.10). In addition, the number of variables to be included was established a priori to avoid over-fitting the model. Finally, we repeated analysis and forced the variables female sex and age (that did not show statistical significance in the univariate analysis) into the model, as they have been associated to patients' preferences for autonomous decision-making in some populations [31].

Missing data were below 1% and applied to patient-reported outcomes (HAQ-DI score and SF-36 score; 2 missing data, each); no imputation was performed.

All statistical analyses were performed using Statistical Package for the Social Sciences version 21.0 (SPSS Chicago IL). A value of p<0.05 was considered statistically significant.

## Ethics

The study was approved by the Internal Review Board of the Institution (Reference number: IRE-3005-19-20-1).

Consecutive patients from the outpatient clinic of the Department of Immunology and Rheumatology were invited to participate; informed consent process was performed in all the patients who agreed, and all of them additionally provided written informed consent.

There were 13 outpatients who participated in the phase 1 of the study (Translation and cultural adaptation to Mexico of the IPAS) who provided extensive feedback. Selected patients gave verbal informed consent that was documented by one of the co-authors who behave as a witness during informed consent process.

Prior to patients´ enrolment, the study was presented to all the rheumatologists assigned to the outpatient clinic during the study period; physicians were invited to fill the IPAS and to provide socio-demographic data. All the physicians agreed, and they gave verbal informed consent as a written informed consent was deemed unnecessary [32].

In all the cases data were de-identified and no personal information was available at any time.

## Results

### Phase 1. Translation and cultural adaptation to Mexico of the IPAS

The process included ten steps, which are summarized in Supplementary Table 2 in S2 Appendix (Please refer to the S2 "Supplementary Table 2 in S2 Appendix. Summary of the cultural adaptation and translation process", S2 Appendix). The Spanish version of the IPAS was subjected to psychometric validation.

### Phase 2. IPAS psychometric validation

**Samples description.** Two samples were used: S-1 for pilot testing and S-2 for psychometric validation. A total of 281 outpatients were invited to participate and 40 declined, primarily due to time constraints, 8 and 32 patients, respectively, from S1 and S2 Appendices.

The 40 patients that integrated S-1 had the following diagnosis: 17 patients (42.5%) had SLE, 12 (30%) had RA, and 3 (7.5%) patients each had Sclerodermia, SV, and PSS, respectively, and the remaining 2 patients (5%) had PAPS. Meanwhile, the 201 patients who integrated S-2 were distributed as follows: 76 patients (37.8%) had SLE, 64 (31.8%) had RA, 14 (7%) had SV, 13 (6.5%) had IM, 11 (5.5%) had Sclerodermia, 8 (4%) had SA, 5 each (2.5%) had PSS and PAPS, respectively, 3 (1.5%) had MCTD, and 2 patients (1%) had adult Still disease.

Patient characteristics are summarized in Table 1. Patients from both samples were primarily middle-aged women, with medium-low socioeconomic status and substantial disease duration, were on immunosuppressive drugs, and their median (IQR) Charlson score was 1 (1–2). Patients from S-2 had religious beliefs (80.6%), economic dependency (54.7%), were living with a partner (48.3%), and a minority had access to the social security system (14.3%). In addition, a minority of the patients had participated in clinical trials (11.9%) or had previous hospitalizations (17.9%). Finally, the median (IQR) pain-VAS was 18 (2–50), HAQ score was 0.5 (0–1.38), and SF-36 score was 60.3 (45.3–76).

**Pilot testing.** IPAS (pre-validation version) was found feasible by the patients included in S-1, as summarized in Table 2. Patients took (median±SD) 6±2 minutes to complete the IPAS. The majority of the patients agreed on instructions and item clarity and IPAS format adequacy. Based on the patients' comments, the IPAS instructions underwent minor modifications.

**Construct validity and IPAS re-structure.** The structure of the IPAS underwent an important modification after factorial analysis, as shown in Fig 1. The 14 items were redistributed into four sub-scales that were renamed as follows: "The patient accepts the doctor´s decision with information" (Sub-scale I), "The patient can challenge the doctor" (sub-scale II), "The doctor should decide" (sub-scale III), and "The doctor has limits" (sub-scale IV). The KMO measure of 0.655 and significant result ($X^2$ = 642.638, p≤0.001) for the Bartlett sphericity test confirmed the adequacy of the sample. A 4-factor structure was extracted, which

**Table 1. Description of the characteristics from the patients that integrated S-1 and S-2.**

| | S-1 N = 40 | S-2 N = 201 |
|---|---|---|
| **Socio-demographic characteristics** | | |
| Female sex | 33 (82.5) | 184 (81.5) |
| Years of age[1] | 49.5 (28–63.3) | 48 (36–59) |
| Years of formal education[1] | 9 (6–16) | 12 (9–17) |
| Medium-low socioeconomic-level | 36 (90) | 187 (93.5) |
| Religious beliefs (primarily Catholics) | NA | 162 (80.6) |
| Economic dependency | NA | 110 (54.7) |
| Living with a partner | NA | 97 (48.3) |
| Access to social security system | NA | 30 (14.3) |
| **Disease-related-characteristics** | | |
| Years of disease duration[1] | 13 (8–19) | 12 (5–20) |
| Years of follow-up at the outpatient clinic[1] | NA | 9 (4–17) |
| Comorbid conditions | NA | 100 (49.8) |
| Charlson score | 1 (1–2) | 1 (1–2) |
| Participation in clinical trials | NA | 24 (11.9) |
| Previous (within one year) hospitalizations | NA | 36 (17.9) |
| Number of hospitalizations within the previous year[1,2] | NA | 1 (1–1) |
| **Patient-reported-outcomes** | | |
| Pain-VAS[1] | NA | 18 (2–50) |
| HAQ-DI score[1,*] | NA | 0.5 (0–1.38) |
| SF-36 score[1,*] | NA | 60.3 (45.3–76) |
| **Disease-related-treatment** | | |
| Immunosuppressive drugs | 37 (92.5) | 198 (98.5) |
| Number of Immunosuppressive drugs/patients[1] | 2 (2–3) | 2 (2–4) |
| Corticosteroids use | 18 (47.5) | 90 (44.8) |

Data described as N° of patients (%) unless otherwise indicated.

[1]Median (IQR).

[2]Restricted to patients with hospitalizations. S = Sample. VAS = Visual Analogue Scale. HAQ-DI = Health Assessment Questionnaire Disability Index. SF-36 = Short Form 36 items.

[*]Two missing data. NA = Not available.

accounted for 56.3% of the total variance. All factors had eigenvalues >1. The factors were equivalent to the four sub-scales.

**Content validity.** Experts generally agreed on item evaluation as summarized in Table 3; item 12 was poorly rated, and after modifications, it was approved by 85% of the experts.

**Table 2. IPAS feasibility (pre-validation version).**

| Feasibility categories | N° (%) of patients |
|---|---|
| Adequate time to fill the questionnaire | 39 (97.5) |
| Incomplete questionnaires[1] | 7 (17.5) |
| Perceived instructions clarity | 33 (82.5) |
| Perceived items clarity | 35 (85.7) |
| Format acceptance | 39 (97.5) |

[1]Five patients referred skipped items and 2 patients needed item explanation.

**Table 3. Experts IPAS items evaluation.**

| Items | Relevance | Adequate wording | Appropriated language and meaning |
|---|---|---|---|
| 2. It is better that the doctor rather than the patient decides which treatment is best. | 100 | 100 | 100 |
| 5. During the conversation, the patient must submit himself/herself with confidence to the expertise of the doctor. | 92 | 100 | 100 |
| 9. If doctor and patient cannot agree on which treatment is best, the doctor should make the treatment decision. | 85 | 92 | 92 |
| 10. The patient should, without much information on the risk involved, confidently undergo a treatment. | 85 | 69 | 77 |
| 12. The doctor can presume that the patient knows that people can die when they receive some treatments. | 46 | 62 | 69 |
| 6. The patient himself/herself must choose between the various treatments. | 85 | 92 | 100 |
| 11. It goes too far when the doctor decides which treatment is best for the patient. | 85 | 92 | 100 |
| 13. If a patient chooses a treatment with more health risks, the doctor should respect this treatment decision. | 92 | 92 | 100 |
| 14. As it concerns the body and life of the patient, the patient should decide. | 92 | 92 | 100 |
| 1. If the patient does not want to receive information about risks, the doctor should respect this. | 100 | 100 | 77 |
| 4. Patients should have the right not to be involved in the decision on the treatment. | 85 | 92 | 92 |
| 8. Patients who become afraid when thinking about the treatment decision should be left in peace by the doctor. | 100 | 100 | 100 |
| 3. The patient has to be informed on all the risks involved in a treatment indication. | 92 | 100 | 100 |
| 7. Before a patient consents to a treatment he/she should receive all information on the risks involved. | 92 | 100 | 100 |

Data presented as % of experts that agree.

**Criterion validity.** We first scored the 201 IPAS and transformed sub-scale scores to a 0–100 scale (5), where 100 corresponded to the highest patient agreement with the items assigned to the sub-scale. We then assigned to each patient a preferred ideal of autonomy based on the highest score from the 4 sub-scales. A total of 165 patients (82.1%) obtained the highest score in one sub-scale, while the 36 patients left (17.9%) scored more than one sub-scale with the highest score. The distribution of patients' ideals of autonomy among the 165 patients with a preferred sub-scale were as follows: 138 patients (83.6%) selected sub-scale I ("The patient accepts doctor´s decision with information"), 6 patients (3.6%) selected sub-scale II ("The patient can challenge the doctor"), 2 patients (1.2%) selected sub-scale III ("The doctor should decide") and 19 patients (11.5%) selected sub-scale IV ("The doctor has limits").

Table 4 summarizes the distribution of patients with a preferred decision-making role according to their answer to the Sutherland question, among the patients classified according their ideal of autonomy; as shown, the patient´s distribution differed between groups (p = 0.024, $X^2$ test).

**Internal consistency and reliability of IPAS.** Results of internal consistency (Cronbach´s α) and reliability/test-retest (ICC and 95% CI) of the IPAS and each IPAS sub-scale are presented in Table 5, which additionally presents floor and ceiling effects. The mean (±SD) of the time between the two measurements in the test-retest analysis was 7.6 (±1.23) days.

## Phase 3. Patient-doctor concordance of autonomy ideal and the association with the patient-doctor relationship

**Description of S-3.** A total of 691 outpatients were invited to participate and 90 declined due mostly to time constraints. The characteristics of the 601 patients who integrated S-3 are

**Table 4. Preferred decision making role among the patients classified according their autonomy ideal.**

| | Sub-scale I, N = 138 | Sub-scale II, N = 6 | Sub-scale III, N = 2 | Sub-scale IV, N = 19 |
|---|---|---|---|---|
| Physician should decide, based on all that is known (N = 29). | 28 | 1 | 0 | 0 |
| Physician should decide strongly taking the patient´s opinion into account (N = 65). | 51 | 3 | 2 | 9 |
| Physician and patient should decide together, based on equity (N = 59). | 52 | 2 | 0 | 5 |
| Patient should decide, strongly taking the physician´s opinion into account (N = 12). | 7 | 0 | 0 | 5 |

No patient selected the Sutherland option: "Patient should decide, based on all that he or she knows or hears about the treatment".

depicted in Table 6 and were similar to those from patients that integrated S-2. In addition, the majority of the patients had pain under control (68.5%), no disability (55.2%), and health-related quality of life out of normal range (73.8%-78.6%), based on published cut-offs [33]. Patients scored high on PDRQ-9, with a median (IQR) of 4.6 (3.4–5) and 30.4% of the patients rated the patient-doctor relationship with the highest score. Finally, the patients´ diagnoses were as follows: 219 patients (36.4%) had SLE, 192 (32%) had RA, 42 (7%) had SV, 23 (3.8%) had IM and PAPS, 25 (4.2%) had Sclerodermia, 28 (4.7%) had SA, 20 each (3.3%) had PSS and MCTD, respectively, and 9 patients (1.5%) had adult Still Disease.

**Patients' and physicians' ideals of autonomy.** There were 497 patients (82.7%) with a preferred ideal of autonomy, while the 104 patients left (17.3%) scored at least two sub-scales with the highest score. Patients from the former group had higher years of formal education (median [IQR]: 12 [9–16] vs. 11 [6.5–16], p = 0.031), lower years of follow-up at the outpatient clinic (median [IQR]: 8.4 [4–16.2] vs. 11.4 [5.1–19.5], p = 0.017) and tended to be less frequently classified with medium-low socioeconomic status (87.7% vs. 94.2%, p = 0.059) than their counterparts.

Among the patients with a preferred autonomy ideal, 415 (83.5%) identified themselves with sub-scale I ("The patient accepts doctor´s decision with information "), 23 (4.6%) with sub-scale II ("The patient can challenge the doctor"), 7 (1.4%) with sub-scale III ("The doctor should decide"), and 52 patients (10.4%) with sub-scale IV ("The doctor has limits"). This distribution was maintained within each rheumatologic disease as summarized in Table 7: 63.2%-100% of the patients from each rheumatic diagnosis identified themselves with sub-scale I, 3.8%-15.8% with sub-scale II, 0.6%-5.9% with sub-scale III, and 0%-18.2% with sub-scale IV.

**Table 5. IPAS internal consistency and reliability/temporal stability.**

| | Cronbach´s α | ICC* | 95% CI* | Floor/ceiling effect (%) |
|---|---|---|---|---|
| IPAS | 0.631 | 0.943 | 0.902–0.967 | 0/0 |
| Sub-scale 1 (4 items), "The patient accepts doctor´s decision with information" | 0.786 | 0.915 | 0.855–0.951 | 2/24.9 |
| Sub-scale 2 (4 items), "The patient can challenge the doctor" | 0.589 | 0.974 | 0.955–0.985 | 1/3 |
| Sub-Scale III (4 items), "The doctor should decide" | 0.514 | 0.973 | 0.953–0.985 | 1/0.5 |
| Sub-scale IV (2 items), "The doctor has limits" | 0.230 | 0.933 | 0.884–0.961 | 1/9 |

ICC = Intraclass Correlation coefficient. CI = Confidence Interval.

*Limited to 50 patients.

**Table 6. Characteristics from patients that integrated S-3.**

| Sociodemographic characteristics | | Disease-related characteristics | |
|---|---|---|---|
| Female sex* | 517 (86) | Disease duration | 11 (5.2–19.4) |
| Years of age | 48.8 (36–59.6) | Disease duration <5 years* | 134 (22.3) |
| Years of formal education | 12 (9–16) | Disease duration of 5–10 years* | 156 (26) |
| Medium-low socioeconomic level* | 534 (88.9) | Disease duration > 10 years* | 311 (51.7) |
| Religious beliefs* | 545 (90.7) | Years of follow-up at the outpatient clinic | 9.3 (4–17) |
| Economic dependency* | 357 (59.4) | Comorbid conditions* | 353 (58.7) |
| Living with a partner* | 305 (50.7) | Charlson score | 1 (1–2) |
| Access to Social Security System* | 105 (17.5) | Research trials participation* | 66 (11) |
| **Patient-reported outcomes** | | Previous hospitalizations* | 81 (13.5) |
| Pain-VAS score | 13 (1–40) | Number of previous hospitalizations[1] | 1 (1–1) |
| Pain-VAS score ≤30 mm* | 411 (68.5) | **Disease-related-treatment** | |
| HAQ-DI score | 0.38 (0–1.13) | Immunossupressive drugs* | 580 (96.5) |
| HAQ-DI ≤0.5* | 332 (55.2) | Number of immunossupressive drugs/patient | 1 (1–2) |
| SF-36 global score | 60.9 (46.3–75.6) | Corticosteroids use* | 268 (44.6) |
| SF-36 physical component ≥79* | 128 (21.4) | **Doctor-patient relationship** | |
| SF-36 emotional component ≥77* | 157 (26.2) | PDRQ-9 score | 4.6 (3.4–5) |
| | | Highest PDRQ-9 score* | 183 (30.4) |

Data presented as median (IQR) as otherwise indicated.

*Number (%) of patients. VAS = Visual Analogue Scale. HAQ = Health Assessment Questionnaire Disability Index.SF-36 = Short Form-36.[1]Limited to patients with previous hospitalizations. PRDQ = Patient-Doctor Relationship Questionnaire.

IPAS was applied to 21 rheumatologists, primarily females (14 [66.7%]), with a median (IQR) age of 37 years (31.5–48), among whom, four physicians (19%) were senior rheumatologists (≥20 years of clinical experience). All of them scored one IPAS sub-scale with the highest score: 15 (71.4%) sub-scale I, four (19%) sub-scale II, and two physicians (9.5%) sub-scale IV.

*Patient-physician concordance of the preferred ideal of autonomy and association with the PDRQ-9 score (primary objective).*

There were 363 patient-doctor encounters with concordance in the preferred ideal of autonomy sub-scale, while 133 encounters did not. Table 8 summarizes the results of the comparison between the global PDRQ-9 score and individual item scores among the encounters with/without concordance and highlights that scores were significantly higher among the encounters with patient-doctor concordance in the preferred ideal of autonomy, but for item seven ("I can talk to my doctor") where a tendency was shown.

**Table 7. Autonomy ideal distribution, among the 497 patients with a preferred autonomy ideal, according to specific rheumatic disease diagnosis.**

| | D-1 N = 180 | D-2 N = 157 | D-3 N = 23 | D-4 N = 33 | D-5 N = 17 | D-6 N = 22 | D-7 N = 19 | D-8 N = 19 | D-9 N = 19 | D-10 N = 8 |
|---|---|---|---|---|---|---|---|---|---|---|
| I | 152 (84.4) | 130 (82.2) | 22 (95.7) | 26 (78.8) | 13 (76.5) | 15 (68.2) | 18 (94.7) | 12 (63.2) | 19 (100) | 8 (100) |
| II | 7 (3.9) | 6 (3.8) | 0 | 2 (6.9) | 1 (5.9) | 3 (13.6) | 1 (5.3) | 3 (15.8) | 0 | 0 |
| III | 1 (0.6) | 3 (1.9) | 1 (4.3) | 0 | 1 (5.9) | 0 | 0 | 1 (5.3) | 0 | 0 |
| IV | 20 (11.1) | 18 (11.5) | 0 | 5 (15.2) | 2 (11.8) | 4 (18.2) | 0 | 3 (15.8) | 0 | 0 |

Data presented as N˚ (%) of patients. D-1 = SLE; D-2 = RA; D-3 = Sclerodermia; D-4 = SV; D-5 = PSS; D-6 = SA; D-7 = IM; D-8 = PAPS; D-9 = MCTD; D-10 = Adult Still disease. I = "The patient accepts doctor´s decision with information"; II = " The patient can challenge the doctor"; III = " The doctor should decide"; IV = " The doctor has limits".

**Table 8. Comparison of global PDRQ-9 score and individual items scores among patient-doctor encounters with/without concordance in the preferred ideal of autonomy.**

| | Patient-doctor encounters with concordance N = 363 | Patient-doctor encounters without concordance N = 133 | p |
|---|---|---|---|
| Global PDRQ-9 score | 4.7 (3.7–5) | 4.1 (3–4.8) | ≤0.001 |
| Item-1 score "My doctor helps me" | 5 (4–5) | 4 (3–5) | ≤0.001 |
| Item-2 score "My doctor has enough time for me" | 4 (3–5) | 4 (3–5) | 0.004 |
| Item-3 score "I trust my doctor" | 5 (4–5) | 5 (3–5) | 0.001 |
| Item-4 score "My doctor understand me" | 5 (3–5) | 4 (3–5) | ≤0.001 |
| Item-5 score "My doctor is dedicated to help me" | 5 (4–5) | 4 (3–5) | 0.002 |
| Item-6 score "My doctor and I agree on the nature of my medical symptoms" | 5 (4–5) | 4 (3–5) | 0.001 |
| Item-7 score "I can talk to my doctor" | 5 (3–5) | 4 (3–5) | 0.094 |
| Item-8 score "I feel pleased with my doctor´s treatment" | 5 (4–5) | 4 (3–5) | ≤0.001 |
| Item-9 score "I find my doctor easily accessible" | 5 (4–5) | 4 (3–5) | ≤0.001 |

PDRQ = Patient Doctor Relationship Questionnaire.

Simple linear regression showed a significant relationship between increased PDRQ-9 global scores and patient-doctor concordance in their ideal of autonomy (β coefficient: 0.363, 95%CI: 0.184–0.541, p ≤0.001, $R^2$ = 0.031). A significant relationship was also found with PDRQ-9 individual item scores (but item 3, p≤0.001 for the items left); the three PDRQ-9 items with the highest increases were those related to patient perceived time dedicated by the doctor, doctor accessibility, and help from the doctor, as summarized in Fig 2.

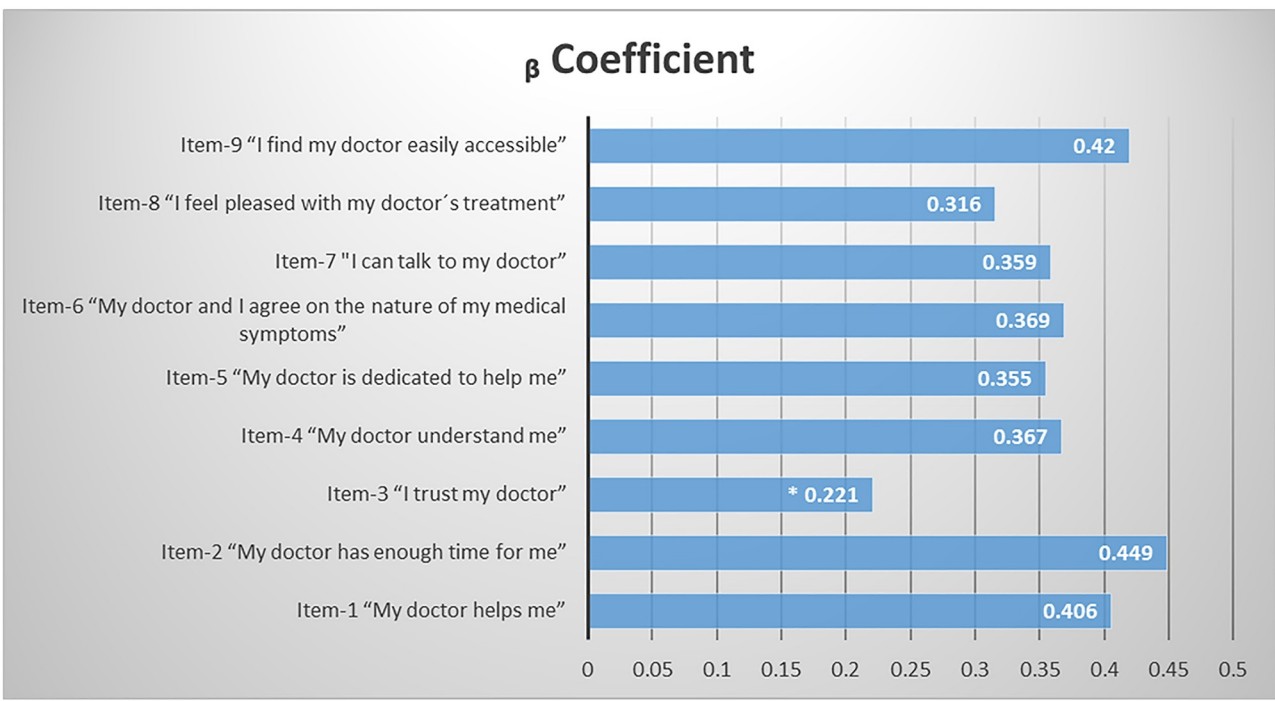

**Fig 2. β coefficients of simple linear regression analysis, to ascertain the magnitude of the relationship between individual PDRQ-9 items scores and patient-doctor encounters with concordance in the ideal of autonomy.** p≤0.001 but * (p = 0.06).

**Factors associated with a paternalistic/physician-centered ideal of patient autonomy.**
The 497 patients from S-3 with a preferred ideal of autonomy were further categorized into two groups: patients with an ideal of autonomy paternalistic (IPAS sub-scale I "The patient accepts doctor´s decision with information" and sub-scale III "The doctor should decide"), and patients with an ideal of autonomy patient-centered (IPAS sub-scale II "The patient can challenge the doctor" and sub-scale IV "Doctor has limits"). There were 422 patients (85.1%) in the first group and 75 (14.9%) in the second group; their characteristics were compared and are summarized in Table 9. Patients from the former group scored the PDRQ-9 higher and a higher percentage of them scored the PDRQ-9 with the highest value; also, they tended to refer more frequently having religious beliefs, living with a partner, and tended to have shorter disease duration.

Finally, the following variables were included in the multiple logistic regression analysis to identify factors associated with a patient preferred ideal of autonomy physician-centered or paternalistic, which was considered the dependent variable: religious beliefs, living with a partner, short disease duration ($\leq$ 5 years), and individual PDRQ-9 item scores, which showed Spearman correlation values of 0.6–0.8 (p$\leq$0.001). Religious beliefs (OR: 2.277, 95% CI: 1.069–4.849, p = 0.033) and a higher PDRQ-9 item eight score ("I feel pleased with my doctor´s treatment", OR: 1.787, 95% CI: 1.425–2.239, p$\leq$0.001) were the only variables associated with a patient ideal of autonomy physician-centered or paternalistic. Same results were obtained when age and female sex were forced into the models.

## Discussion

The present study revealed that the concordance between rheumatologists and their patients about their preferred ideal of autonomy was associated with a better patient-doctor relationship, in Mexican outpatients with representative rheumatic diseases. The magnitude of the relationship between patient-doctor encounters with concordance in the ideal of autonomy and PDRQ-9 individual item scores showed variations; higher increases were evident for PDRQ-9 items related to patients' perceived time dedicated by the doctor, doctor accessibility, and help from the doctor. The study also revealed that in our population, patients and rheumatologists preferred ideal of autonomy were primarily physician-centered or paternalistic; having religious beliefs and higher levels of patient satisfaction with the treatment prescribed were associated with an paternalistic ideal of autonomy. Finally, the translated and culturally adapted (to Mexico) IPAS factor structure differed from that of the original instrument, although IPAS was found valid, with mild internal consistency, good to excellent temporal stability, and feasible.

Our primary objective focused on the patient-doctor relationship, a complex dynamic and multidisciplinary phenomenon, culturally influenced, contextually driven, and difficult to measure in its overall extension. We used the PDRQ-9 items Hispanic version, which has shown adequate psychometric properties to assess the construct in Spanish speaking patients [22]. In the clinical context of ambulatory care medicine (but not limited to), a good patient-doctor relationship is highly valuable in itself. In addition, there is published literature that emphasizes the positive impact of the patient-doctor communication/relationship on health outcomes [34], which can be extended to the field of rheumatic diseases [13, 16, 17, 19, 35]. Traditionally, the construction of patient-doctor relation is a relation of trust, guided by concern for the patient´s best interest, from professionals with relevant skills and knowledge [36]. Nonetheless, this trust-based model seems increasingly obsolete as contemporary relations between patients and doctors are constrained, formalized, and regulated in many ways [36]. Doctors find themselves pressed to be accountable rather than to be communicative, to

**Table 9. Comparison of characteristics from patients with ideal of autonomy physician-centered (paternalistic) and those with ideal of autonomy patient-centered.**

| | Patients with ideal of autonomy paternalistic, N = 422 | Patients with ideal of autonomy patient-centered, N = 75 | p |
|---|---|---|---|
| **Socio-demographic characteristics** | | | |
| Female sex | 368 (87.2) | 61 (81.3) | 0.2 |
| Years of age[1] | 47.9 (36.2–59.5) | 50.8 (33.8–58.9) | 0.854 |
| Years of formal education[1] | 12 (9–16) | 12 (9–16) | 0.757 |
| Medium-low socioeconomic-level | 369 (87.4) | 67 (89.3) | 0.848 |
| Religious beliefs (Catholics) | 387 (91.7) | 64 (85.3) | 0.086 |
| Economic dependency | 248 (58.8) | 48 (64) | 0.445 |
| Living with a partner | 221 (52.4) | 30 (40) | 0.06 |
| Access to Social Security System | 78 (18.5) | 10 (13.3) | 0.327 |
| **Disease-related characteristics** | | | |
| Disease duration[1] | 10.8 (5.2–19.3) | 10 (4–18.2) | 0.190 |
| Disease duration <5 years | 94 (22.3) | 24 (32) | 0.077 |
| Disease duration of 5–10 years | 111 (26.3) | 20 (26.7) | 1 |
| Disease duration > 10 years | 217 (51.4) | 31 (41.3) | 0.132 |
| Years of follow-up at the outpatient clinic[1] | 9.2 (4.2–16.3) | 7.2 (2.7–15.2) | 0.142 |
| Comorbid conditions | 255 (60.4) | 41 (54.7) | 0.373 |
| Charlson score[1] | 1 (1–2) | 1 (1–2) | 0.130 |
| Research trials participation | 45 (10.7) | 5 (6.7) | 0.404 |
| Previous hospitalizations | 58 (13.7) | 9 (12) | 0.854 |
| Number of previous hospitalizations[1, 2] | 1 (1–1) | 1 (1–1) | 0.488 |
| **Patient-reported-oucomes** | | | |
| Pain-VAS score[1] | 14 (1–40.5) | 15 (0–47) | 0.734 |
| Pain-VAS score ≤30 mm | 290 (68.9) | 51 (68) | 0.893 |
| HAQ-DI score[1] | 0.375 (0–1.125) | 0.375 (0–1.375) | 0.819 |
| HAQ-DI ≤0.5 | 234 (55.5) | 40 (53.3) | 0.801 |
| SF-36 global score [1] | 60.5 (46.5–75.2) | 55.6 (45.2–77.2) | 0.824 |
| SF-36 physical component ≥79 | 86 (20.4) | 17 (22.7) | 0.645 |
| SF-36 emotional component ≥77 | 102 (24.2) | 22 (29.3) | 0.383 |
| **Disease-related-treatment** | | | |
| Immunossupressive drugs | 406 (96.2) | 73 (97.3) | 1 |
| Number of immunossupressive drugs/ patient[1] | 1 (1–2) | 1 (1–2) | 0.748 |
| Corticosteroids use | 187 (44.3) | 34 (45.3) | 0.9 |
| **Doctor-patient relationship** | | | |
| PDRQ-9 score[1] | 4.5 (3.6–5) | 3.6 (2.8–4.6) | 0.000 |
| Highest PDRQ-9 score | 132 (31.3) | 12 (16) | 0.008 |

Data presented as number (%) unless [1]median (IQR).[2]Restricted to patients with the characteristic. VAS = Visual Analogue Scale. HAQ = Health Assessment Questionnaire Disability Index.SF-36 = Short Form-36.[1]Limited to patients with previous hospitalizations. PRDQ = Patient-Doctor Relationship Questionnaire.

conform to regulations rather than to enter relationships of trust [36]. Meanwhile, most contemporary accounts of autonomy see it as a form of independence, and independence is relational (you are independent of someone), as it is a medical encounter [36]. Human encounters involve individuals, who are never fully formed, but are always dynamically in the process of development [7]; this development should be enriched by both participants and directed to build solid relationships.

The primary conclusion of the study was that concordance between rheumatologists and their patients about their ideal of autonomy was associated with a better patient-doctor relationship. We considered that being able to identify one´s choice should be a prerequisite for true autonomy, and this assumption forced us to limit our analysis to patients with one preferred ideal of autonomy (82.1% of the patients). It is worth mentioning that although the IPAS focuses on ideals of autonomy, it may be difficult for patients and physicians to rely on hypothetical situations of ideally autonomous action or choice; rather, individuals must identify specific concrete conditions or features that contribute or define an action of choice as autonomous [7]. The practical pursuit of autonomy varies depending on the context and the different moral agents involved. Given that our target population consisted of patients with a specific rheumatic disease, attending an outpatient clinic that was served by 21 primary rheumatologists, and with each patient having a primary rheumatologist assigned, these conditions integrated the contextual framework necessary to take some distance from the incompleteness and uncertainty that the "actual autonomy" phenomenon presents when compared to ideal autonomy [7]. Our primary result, largely unknown, which highlights the relevance of patient-doctor concordance (in their ideal of autonomy), may be considered an extension of similar published findings related to other domains of the medical encounter. Discordance between expectations of patients and their physicians occurs when each participant assigns different values to a health trait and has the potential to limit health care [37, 38]. In addition, the mutual recognition of problems faced by rheumatic patients and their rheumatologists has been associated with more reported clinical improvements in subsequent visits [13]. Patients recognized that advice and information that are not in accordance with their experience and understanding are a source of stress, while negotiations that are mutually acceptable are strategies that they perceive as more supportive [35]. In addition, patients seeing physicians of the same ethnic background as themselves rated their physicians as more participatory, which has been explained by patients and physicians sharing cultural beliefs, values, and experiences in society, allowing them to communicate more effectively and to feel more comfortable with one another [39]. Finally, RA patients felt that physician review of the patient-reported outcomes obtained at routine visits made the visit more patient centered (instead of paternalistic), improved shared decision-making, resulting in higher satisfaction and treatment confidence [40]. These studies highlight the association between patient-doctor concordance and patient´s perceived health-related benefits; although, strategies to enhance patients' feelings of being understood (which may be considered a surrogate for patient-doctor concordance) might vary depending on the patient´s preference for decision-making (which might be considered a surrogate for autonomy principle) [20]. Finally, we observed differences in the magnitude of the association between the impact of patient-doctor concordance on their ideal of autonomy and individual PDRQ-9 items scores; associations were statistically significant but for item three, that assessed patient´s trust in their primary physician; this result could be explained by the inherent complexity of trust, which is a distinct construct in itself [41], which has been associated with racial, cultural, socio-demographic, and disease-related variables [41, 42, 43]. The other items assessed more concrete aspects of the patient-doctor relationship.

The second relevant result from our study revealed that the majority of Mexican patients with rheumatic diseases and their attending rheumatologists, had an ideal of autonomy that was paternalistic, although receiving information was still important for these patients. Paternalism has been defined as the use of power and authority by an institution or a person over another person, to provide benefit or prevent harm, in which the benefit and harm are defined by the authority; this behavior is carried out with only the implicit consent of the person acted upon, which may be a patient [43]. Paternalism has been described as a frequent attitude among Mexican physicians [44, 45], particularly in (but not limited to) the context of the public

health system. In addition, an evaluation of physicians' communication styles suggests that physicians (not limited to those working in the Latin-American region) use a predominantly doctor-centered approach to medical interviews [19]. From the patient´s perspective, paternalism was not objected to by Mexican patients with fibromyalgia [45]; also, Mexican patients with RA, SLE, and other rheumatic diseases were less likely to desire or undertake an active role at the time of their consultation [12], which is in line with our results, and has been replicated in Canadian patients with rheumatic diseases [10]. Even more, Thompson et al [46] have recently suggested that paternalism may be a more highly valued ethic in some cultural contexts, such as the Latino patients from USA, in whom the sole emphasis on patient autonomy could potentially have negative consequences such as the alienation of Latino patients and lower rates of treatment adherence. Also, it should be highlighted that our population was integrated primarily by females, and females (with cancer) had been found to prefer physician control [47]. Paternalism is typical of most medical encounters that involve patients with chronic diseases and has been related to a "sick role" adopted by the patients [11]. Patients are highly vulnerable to others, and highly dependent on doctor's action and competence [36]. This may be particularly true in high-power-distance cultural communities such as Mexico [11]. As O´Neill has pointed out, a robust conception of autonomy may be unachievable for most patients, who are typically asked to choose from a menu of treatment strategies that we (their rheumatologists) have composed and described in simplified terms; it is probable that a considerable relief to the patients is that they are not asked to master much in the way of individual (and liberal concept of) autonomy [36]. We are in accordance with O´ Neill that the older trust-centered model of the patient-doctor relationship should be revisited and leave obscurantism.

In our population, having religious beliefs and a higher patient´s satisfaction with the treatment prescribed were associated with a paternalistic patient ideal of autonomy. Individual autonomy is generally depicted as a capacity or trait that individuals may have to a greater or lesser degree, which they will manifest by acting independently in the right and appropriate way [36]. The central difficulty for many current accounts of individual autonomy is that its proponents also take a naturalistic view of human action [36]. A naturalistic view of human action is based on natural states and events, in particular by desires and beliefs [36]. Religious beliefs, spirituality, and existential concerns are part of human beings and have also become a major component of health-related quality of life [48]. They are increasingly recognized to have value in clinical care, and proposed to be more openly discussed and taught [49, 50]; meanwhile, spiritual needs may not be a priority for health professionals, relative to more tangible issues [51]. In addition, in some cultures, including ours, a lack of respect toward spirituality and religion might generate a lack of trust in the physician (which is part of the patient-doctor relationship), and consequently non-adherence by patients to treatment schemes [44]. Finally, higher patient satisfaction with the treatment prescribed was also associated with a patient's ideal of autonomy that is physician-centered. Although no causal relationship could be established, this finding questions the negative label given to passive patients with rheumatic diseases in the literature [43], which might be culturally nuanced. Satisfaction with treatment is one of the most relevant aspects of patient-doctor encounters, particularly in the clinical context of rheumatic diseases, as it has a positive impact on adherence to treatment strategies and ultimately in patients outcomes.

Final results that deserve a brief discussion is that the translated and culturally adapted (to Mexico) IPAS factor structure differed from that of the original instrument, although IPAS was found valid, with limited internal consistency, good to excellent temporal stability and feasible. The structure of the IPAS underwent an important modification from the original Dutch scale and may be explained by the autonomy construct, which is developmentally and socially conditioned, so that determinate expressions of autonomy will be unique and contextually

situated [7]. In addition, ideals of autonomy differ by culture, but perceptions of aspects as normatively inherent to autonomy may also be different [5]. A four-factor model was appropriate and together explained 56.3% of the total variance [52]. The Spanish version of the original instrument showed adequate psychometric properties in terms of construct, content, criteria validity, and reliability, which was evaluated with internal consistency and test-retest, as recommended [26]. Nonetheless, internal consistency was low and could be related to the inherent complexity of the construct evaluated. The IPAS was feasible based on patients' evaluation and suitable for low-literacy patients. Different samples of consecutive outpatients with rheumatic diseases, which were representative of real-world outpatients attending a tertiary care level center, were used for analysis, and accordingly we consider our results could be generalized to populations of patients with similar characteristics.

Some limitations of the study need to be addressed. First, the study was conducted at a single academic center in a metropolitan area, where most of the patients (but not trainees) had an established relationship with their primary rheumatologist. Second, patients had different rheumatic diseases, although others highly prevalent such as osteoarthritis were not considered; in addition, our population was primarily conformed by SLE and RA patients, probably due to local reference bias; accordingly, it might be argued that results cannot be generalized to lesser represented rheumatic diseases. We reclassified patients into 4 conditions [53] namely "RA and other synovial disorders" which included RA and adult Still disease patients, "Connective Tissue Disorders" which included patients with SLE, Sclerodermia, PSS, IM, PAPS and with MCTD, "Vasculitides" and "Spondyloarthritis"; we repeated analysis and confirmed that our primary result was consistent across different rheumatic diseases (Please refer to the S3 "Supplementary Table 3 in S3 Appendix. Simple linear regression analysis, to ascertain the magnitude of the relationship between PDRQ-9 score and patient-doctor encounters with concordance in the ideal of autonomy, in 4 rheumatic conditions", S3 Appendix). Third, the majority of our patients were women, which reflects worldwide female preponderance in rheumatic diseases, particularly in SLE and RA; accordingly, results might not be generalized to male patients. Fourth, the findings may be specific to patients and physicians from the Latin-American region due to the importance of the cultural context on autonomy and doctor-patient relationship constructs. Fifth, the IPAS was applied to physicians, although it was not validated in such a population. Sixth, the cross-sectional nature of our study limits the ability to make inferences about causal relationships. Finally, the doctor-patient relationship was assessed through a questionnaire that was limited to the patient´s perspective.

## Conclusions

Patients and physicians have different views on health problems and (unnoticed) differences may impact the patient-doctor relationship. Through communication and commitment, which are fundamental to human relationships, differences can be offset. Patients and their doctors can work together, share, and understand each other's values, ideals, and principles, to build a solid (patient-doctor) relationship, which is the most valuable outcome. Today, the mechanistic view of patients limited to the medical model is no longer satisfactory. Patients and physicians should recognize the value of spiritual beliefs in health-related outcomes. Finally, there are patients with rheumatic diseases who do not seek the level of involvement that the principialist bioethics literature suggests they should; they are willing to leave much of the responsibility for decision-making to their primary rheumatologist, which is in itself an autonomous decision; this paternalistic ideal of autonomy is associated with better aspects of the doctor-patient relationship, enabling the care model of paternalism to keep a significant degree of validity in our community.

## Supporting information

**S1 Appendix. Supplementary Table 1.** Consecutive steps/criteria for item´s reduction N˚ (%) of patients with at least one visit to the outpatient clinic with the 10 most frequent diagnosis specified.
(PDF)

**S2 Appendix. Supplementary Table 2.** Summary of the cultural adaptation and translation process.
(PDF)

**S3 Appendix. Supplementary Table 3.** Simple linear regression analysis, to ascertain the magnitude of the relationship between PDRQ-9 score and patient-doctor encounters with concordance in the ideal of autonomy, in 4 rheumatic conditions.
(PDF)

## Acknowledgments

The authors acknowledge physicians from the Department of Immunology and Rheumatology of the Instituto Nacional de Ciencias Médicas y Nutrición Salvador Zubirán (Ana Barrera Vargas, Hilda Esther Fragoso Loyo, Carlos Edmundo García Padilla, Gabriela Aurora Hernández Molina, Andrea Hinojosa Azaola, Juan de Jesús Jakez Ocampo, Francisco Javier Merayo Chalico, Tatiana Sofía Rodríguez Reyna, Juanita Romero Díaz, José Jiram Torres Ruíz, Marina Rull Gabayet, Lilian Guadalupe Cano Cruz, Héctor Alejandro Culebro Bermejo, Maricruz Quintana, Rosa Carina Soto Fajardo, Guillermo Arturo Guaracha Basáñez, Susy Marcela Sánchez Cubias, Graciela Sandoval Flores, Daniel Giovanni Vázquez López, Luis Guillermo Llorente Peters) and expert Committee members (Everardo Álvarez-Hernández, César Alejandro Arce Salinas, Miguel Ángel Saavedra Salinas, Luis Manuel Amezcua Guerra, Judith López Zepeda, Mónica Vázquez del Mercado, Mario H Cardiel Ríos, Juan Sierra Madero, Mariana Checa Pastrana, Victoria Angélica Torres Añorve, Sara Elena Villanueva Sáenz, Claudia del Socorro Villanueva Sáenz, Stéphanie Derive, María de la Luz Casas Martínez, Luis Felipe Flores- Suárez).

## Author Contributions

**Conceptualization:** Virginia Pascual-Ramos, Christiaan Molewijk Albert, Gregorio Tomás Obrador, Evandro Agazzi.

**Formal analysis:** Virginia Pascual-Ramos, Irazú Contreras-Yáñez.

**Investigation:** Irazú Contreras-Yáñez, Ana Belén Ortiz-Haro, Gregorio Tomás Obrador, Evandro Agazzi.

**Methodology:** Virginia Pascual-Ramos, Irazú Contreras-Yáñez.

**Supervision:** Virginia Pascual-Ramos, Christiaan Molewijk Albert, Gregorio Tomás Obrador, Evandro Agazzi.

**Validation:** Virginia Pascual-Ramos, Irazú Contreras-Yáñez, Ana Belén Ortiz-Haro, Christiaan Molewijk Albert, Gregorio Tomás Obrador, Evandro Agazzi.

**Visualization:** Virginia Pascual-Ramos, Evandro Agazzi.

**Writing – original draft:** Virginia Pascual-Ramos.

**Writing – review & editing:** Virginia Pascual-Ramos, Irazú Contreras-Yáñez, Ana Belén Ortiz-Haro, Christiaan Molewijk Albert, Gregorio Tomás Obrador, Evandro Agazzi.

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
