## [Decision Letter · Decision Letter 0]

15 Jul 2020

PONE-D-20-11971

Concordance among patients and physicians about their ideal of autonomy impacts the patient-doctor relationship: a cross-sectional study of Mexican patients with rheumatic diseases.

PLOS ONE

Dear Dr. Pascual-Ramos,

Thank you for submitting your manuscript to PLOS ONE. After careful consideration, we feel that it has merit but does not fully meet PLOS ONE’s publication criteria as it currently stands. Therefore, we invite you to submit a revised version of the manuscript that addresses the points raised during the review process.

We look forward to receiving your revised manuscript.

Kind regards,

Luca Navarini

Academic Editor

PLOS ONE

2. Please provide additional details regarding participant consent. In the ethics statement in the Methods and online submission information, please ensure that you have specified how verbal consent was documented and witnessed.

3. Please amend your authorship list in your manuscript file to include author Gregorio Tomás Obrador.

Reviewers' comments:

Reviewer's Responses to Questions

**Comments to the Author**

1. Is the manuscript technically sound, and do the data support the conclusions?

Reviewer #1: Yes

Reviewer #2: Yes

2. Has the statistical analysis been performed appropriately and rigorously? 

Reviewer #1: Yes

Reviewer #2: Yes

3. Have the authors made all data underlying the findings in their manuscript fully available?

Reviewer #1: Yes

Reviewer #2: Yes

4. Is the manuscript presented in an intelligible fashion and written in standard English?

Reviewer #1: Yes

Reviewer #2: Yes

5. Review Comments to the Author

Reviewer #1: In this interesting article Pascual Ramos and colleagues investigated the concordance among patients and physicians about their ideal of autonomy impacts the patient-doctor relationship. The article is well written and provides interesting insights in the topic. My major concerns relate to:

-Gender bias: Almost all patients included in all phases of the study are females and although this reflects the prevalence of the included diseases (e.g. SLE, RA), this necessarily hampers the generalizability of the results. This in fact prevents the assessment of whether gender is a factor that should be considered in regression models.

-Imbalance of disease types: SLE and RA account for more that half of the sample so again the question remains on how these results can be generalised to all other diseases that are poorly represented in the sample. Perhaps it would make sense to pool patients by disease groups (e.g. inflammatory arthritis vs connective tissue diseases, ideally by including additional patients to balance the numbers). This would also allow to unmask disease-related differences.

Reviewer #2: The authors found that concordance of autonomy ideal among patients and their rheumatologists positively impacts on the patient-doctor relationship.

The study also revealed that Mexican patients and rheumatologists ideal of autonomy was primarly physician-centered or paternalistic.

This is an interesting study that reveals that the concordance about ideal autonomy improves patient - doctor relationship in Mexican patients with rheumatic diseases.

In literature is reported that patients’ preferences and autonomous decision-making are influenced by demographic variables (with younger ,better educated patients and women being quite consistently found to prefer a more active role in decision making), their experience of illness and medical care, their diagnosis and health status , the type of decision they need to make, the amount of knowledge they have acquired about their condition, their attitude towards involvement, and the interactions and relationships they experience with health professionals. Their preferences are likely to develop over time as they gain experience and may change at different stages of their illness .

I think that this study enriches the knowledgement about clinical decision making in a specific geographic area.

I appreciated methodological steps and interdisciplinary collaboration.

I have not any major or minor remarks.

6. PLOS authors have the option to publish the peer review history of their article (what does this mean?). If published, this will include your full peer review and any attached files.

Reviewer #1: No

Reviewer #2: **Yes: **Palma Scolieri

---

## [Author Response · Author response to Decision Letter 0]

22 Jul 2020

RESPONSE TO REVIEWERS

JOURNAL REQUIREMENTS

 Response: We have revised PLOS ONE´s manuscript style requirements. 

2. Please provide additional details regarding participant consent. In the ethics statement in the Methods and online submission information, please ensure that you have specified how verbal consent was documented and witnessed.

Response: We have updated the ethics statement in Methods section and in the online submission.

3. Please amend your authorship list in your manuscript file to include author Gregorio Tomás Obrador.

Response: We have amended the authorship list. 

COMMENTS TO THE AUTHOR

Reviewer #1: In this interesting article Pascual Ramos and colleagues investigated the concordance among patients and physicians about their ideal of autonomy impacts the patient-doctor relationship. The article is well written and provides interesting insights in the topic. 

Response: We appreciate the reviewer´s comments. 

My major concerns relate to:

-Gender bias: Almost all patients included in all phases of the study are females and although this reflects the prevalence of the included diseases (e.g. SLE, RA), this necessarily hampers the generalizability of the results. This in fact prevents the assessment of whether gender is a factor that should be considered in regression models.

Response: We agree with the reviewer and have added as a limitation of the study. In addition, we have performed two different analysis, and updated the corresponding paragraph; in the first one, only variables with statistical significance in the univariate analysis were included into the models; in the second analysis, female sex and age were additionally forced into the model, as both had been associated to share-decision models in some populations (an additional reference is provided, number 31); importantly, with both analysis same results were obtained. 

-Imbalance of disease types: SLE and RA account for more that half of the sample so again the question remains on how these results can be generalised to all other diseases that are poorly represented in the sample. Perhaps it would make sense to pool patients by disease groups (e.g. inflammatory arthritis vs connective tissue diseases, ideally by including additional patients to balance the numbers). This would also allow to unmask disease-related differences.

Response: We agree with the reviewer; our population is highly represented by SLE and RA patients (probably due to local reference bias); we have added in the manuscript table 7, which summarizes a similar distribution of preferred autonomy ideal among the different rheumatic diseases. In addition, we have addressed in the limitation section, that the imbalance observed, prevents results generalization; we also propose a patient´s redistribution into 4 categories (a reference is provided) and show the results in supplementary table 3. 

Reviewer #2: The authors found that concordance of autonomy ideal among patients and their rheumatologists positively impacts on the patient-doctor relationship.

The study also revealed that Mexican patients and rheumatologists ideal of autonomy was primarily physician-centered or paternalistic.

This is an interesting study that reveals that the concordance about ideal autonomy improves patient - doctor relationship in Mexican patients with rheumatic diseases.

In literature is reported that patients’ preferences and autonomous decision-making are influenced by demographic variables (with younger ,better educated patients and women being quite consistently found to prefer a more active role in decision making), their experience of illness and medical care, their diagnosis and health status, the type of decision they need to make, the amount of knowledge they have acquired about their condition, their attitude towards involvement, and the interactions and relationships they experience with health professionals. Their preferences are likely to develop over time as they gain experience and may change at different stages of their illness.

I think that this study enriches the knowledgment about clinical decision making in a specific geographic area.

I appreciated methodological steps and interdisciplinary collaboration.

I have not any major or minor remarks

Response: We highly appreciate the reviewer´s comments.

---

## [Decision Letter · Decision Letter 1]

6 Oct 2020

Concordance among patients and physicians about their ideal of autonomy impacts the patient-doctor relationship: a cross-sectional study of Mexican patients with rheumatic diseases.

PONE-D-20-11971R1

Dear Dr. Pascual-Ramos,

We’re pleased to inform you that your manuscript has been judged scientifically suitable for publication and will be formally accepted for publication once it meets all outstanding technical requirements.

Kind regards,

Luca Navarini

Academic Editor

PLOS ONE

Additional Editor Comments (optional):

Reviewers' comments:

Reviewer's Responses to Questions

**Comments to the Author**

1. If the authors have adequately addressed your comments raised in a previous round of review and you feel that this manuscript is now acceptable for publication, you may indicate that here to bypass the “Comments to the Author” section, enter your conflict of interest statement in the “Confidential to Editor” section, and submit your "Accept" recommendation.

Reviewer #1: All comments have been addressed

Reviewer #2: (No Response)

2. Is the manuscript technically sound, and do the data support the conclusions?

Reviewer #1: Yes

Reviewer #2: Yes

3. Has the statistical analysis been performed appropriately and rigorously? 

Reviewer #1: Yes

Reviewer #2: Yes

4. Have the authors made all data underlying the findings in their manuscript fully available?

Reviewer #1: Yes

Reviewer #2: Yes

5. Is the manuscript presented in an intelligible fashion and written in standard English?

Reviewer #1: Yes

Reviewer #2: Yes

6. Review Comments to the Author

Reviewer #1: Authors properly addressed all issues that have been raised. I have no further comments on the revised version of the manuscript.

Reviewer #2: I confirm my positive feedback about this paper. The article is well constructed and

may be accepted for publication.

Please note that at line 124 it needs add p to “suplementary”

7. PLOS authors have the option to publish the peer review history of their article (what does this mean?). If published, this will include your full peer review and any attached files.

Reviewer #1: No

Reviewer #2: **Yes: **PALMA SCOLIERI

---

## [Editor Report · Acceptance letter]

12 Oct 2020

PONE-D-20-11971R1 

Concordance among patients and physicians about their ideal of autonomy impacts the patient-doctor relationship: a cross-sectional study of Mexican patients with rheumatic diseases. 

Dear Dr. Pascual-Ramos:

I'm pleased to inform you that your manuscript has been deemed suitable for publication in PLOS ONE. Congratulations! Your manuscript is now with our production department. 

Kind regards, 

on behalf of

Dr. Luca Navarini 

Academic Editor

PLOS ONE